# Lasianosides A–E: New Iridoid Glucosides from the Leaves of *Lasianthus verticillatus* (Lour.) Merr. and Their Antioxidant Activity

**DOI:** 10.3390/molecules24213995

**Published:** 2019-11-05

**Authors:** Gadah Abdulaziz Al-Hamoud, Raha Saud Orfali, Shagufta Perveen, Kenta Mizuno, Yoshio Takeda, Tatsuo Nehira, Kazuma Masuda, Sachiko Sugimoto, Yoshi Yamano, Hideaki Otsuka, Katsuyoshi Matsunami

**Affiliations:** 1Graduate School of Biomedical and Health Sciences, Hiroshima University, 1-2-3 Kasumi, Minami-ku, Hiroshima 734-8553, Japan; galhamoud@ksu.edu.sa (G.A.A.-H.); ssugimot@hiroshima-u.ac.jp (S.S.); yamano@hiroshima-u.ac.jp (Y.Y.); otsuka-h@yasuda-u.ac.jp (H.O.); 2Department of Pharmacognosy, College of Pharmacy, King Saud University, Riyadh 11495, Saudi Arabia; Shagufta792000@yahoo.com; 3Faculty of Integrated Arts and Sciences, The University of Tokushima, 1-1 Minamijosanjima-Cho, Tokushima 770-8502, Japan; tokushis1@yahoo.co.jp (K.M.); takeda@ias.tokushima-u.ac.jp (Y.T.); 4Graduate School of Integrated Arts and Sciences, Hiroshima University, 1-7-1 Kagamiyama, Higashi-Hiroshima 739-8521, Japan; tnehira@hiroshima-u.ac.jp (T.N.); b163780@hiroshima-u.ac.jp (K.M.); 5Faculty of Pharmacy, Yasuda Women’s University, 6-13-1 Yasuhigashi, Asaminami-ku, Hiroshima 731-0153, Japan

**Keywords:** *Lasianthus trichophlebus*, Rubiaceae, DPPH, radical scavenging activity, cytotoxicity

## Abstract

The genus *Lasianthus* (Rubiaceae) consists of approximately 180 species, of which the greatest species diversity is found in tropical Asia. Some of the *Lasianthus* species have been used in folk medicine to treat tinnitus, arthritis, fever, and bleeding. *Lasianthus verticillatus* (Lour.) Merr. (Syn. *Lasianthus trichophlebus* auct. non Hemsl.) is a shrub, branchlets terete about 1.5–3 m in height. This paper studies the chemical composition of the leaves of *L. verticillatus* for the first time, which resulted in the isolation of five undescribed iridoid glucosides, lasianosides A–E (**1**–**5**), together with three known compounds (**6**–**8**). The undescribed structures of isolated compounds (**1**–**5**) were characterized by physical and spectroscopic data analyses, including one-dimensional (1D) and two-dimensional (2D) NMR, IR, UV, and high-resolution electrospray ionization mass spectra (HR-ESI-MS). Furthermore, the electronic circular dichroism data determined the absolute configurations of the new compounds. The free radical scavenging properties of isolated compounds was assessed by 1,1-diphenyl-2-picrylhydrazyl (DPPH) radical scavenging assay, and their cytotoxicity was assessed toward human lung cancer cell line A549 by the 3-(4,5-dimethylthiazol-2-yl)-2,5-diphenyltetrazolium bromide (MTT) method. Among the isolated compounds, **3** and **4** displayed potent radical scavenging activities with IC_50_ values of 30.2 ± 1.8 and 32.0 ± 1.2 µM, which were comparable to that of Trolox (29.2 ± 0.39 µM), respectively, while **5** possessed moderate activity with an IC_50_ value of 46.4 ± 2.3 µM. None of the isolated compounds exerted cytotoxicity against human cell line A549. As a result, lasianosides C, D, and E have the potential to be non-toxic safe antioxidant agents.

## 1. Introduction

Reactive oxygen species (ROS) are derived from exogenous (environmental pollution, cigarette smoke, UV irradiation, and toxic chemicals) and intracellular (mitochondrial energy generation) sources. The excessive ROS accumulation leads to oxidative stress through the reaction of ROS with biomolecules such as cell membrane lipids, proteins, and DNA, which causes a variety of chronic and degenerative disorders including cataract, rheumatoid arthritis, cancer, cardiovascular diseases, neurodegenerative disorders, and also aging. Therefore, a supplementation of exogenous antioxidants as free radical scavengers is thought to be an effective measure for preventing and repairing the damages caused by ROS [1].

Lung cancer is one of the leading causes of cancerous death for both the male and female in the world. Treatment options for lung cancer include surgery, chemotherapy, and radiation therapy. However, the prognosis is still unsatisfactory, probably because of its high invasion and metastatic activity to various organs (e.g., brain, lymph nodes, bones, and liver), the ability to evade apoptosis, the resistance to antitumor drugs, and the severe side effects of antitumor drugs and radiotherapy themselves. Therefore, there is an increased and continued demand for new agents to treat and prevent this life-threatening disease [2].

A great number of natural bioactive compounds have been isolated from various natural sources and have contributed to the development of more than 60% of current clinical drugs [3]. Thus, we still rely on Mother Nature, including plants’ secondary metabolites for the discovery of seed and lead chemicals in serendipitous way. Plants of the genus *Lasianthus* (Rubiaceae) are distributed in Asia, Africa, America, and Australia, with about 180 species in total. Most of the species are found in tropical Asia, while only one species exists in Australia [4,5]. Some of the species have been used in folk medicine to treat tinnitus, arthritis, fever, and bleeding [6,7,8,9]. Previous chemical studies on various species of genus *Lasianthus* have resulted in the isolation of iridoids with diverse skeletons, especially asperuloside and deacetyl asperulosidic acid, in addition to bis-iridoid glucosides, anthraquinones, megastigmane glucosides, and terpenes [8,10,11,12,13,14,15]. *L. verticillatus* (Lour.) Merr. (Syn. *L. trichophlebus* auct. non Hemsl.) is a shrub, branchlets terete of about 1.5–3 m in height [16]. *L. verticillatus* has been used as a traditional, analeptic, or restorative medicine in Northern Thailand and India [17]. Recently, Napiroon et al. reported thin-layer chromatography (TLC) and HPLC analyses of the lipophilic fraction of the leaves of *L. verticillatus* to reveal the presence of terpenoids, alkaloids, phenolic compounds, and coumarines by the reaction color of spraying reagents. The seed germination and seedling growth of weeds and anti-fungal activity against a plant pathogen, *Colletotrichum gloeosporioides*, were reported. However, a detailed chemical analysis of the secondary metabolites and their biological activities has not yet been performed. This study is the first report on the chemical constituents of the polar fraction (1-BuOH) of the leaves of *L. verticillatus*, and afforded five new iridoid glucosides, lasianosides A–E (**1**–**5**), together with three known compounds, an iridoid (**6**) (compound **2** in Takeda et al., 2002 [11], designated hereafter as lasianol by the author, YT), deacetyl daphylloside (**7**), and daphylloside (**8**) [18] (Figure 1). This paper deals with the structural elucidation of isolated compounds including the absolute configuration of new compounds, and the evaluation of their antioxidant activities by 1,1-diphenyl-2-picrylhydrazyl (DPPH) free radical scavenging assay, and cytotoxic activity against human lung cancer cell line (A549) that is frequently used for the study of lung cancer and the initial drug screening, by 3-(4,5-dimethylthiazol-2-yl)-2,5-diphenyltetrazolium bromide (MTT) assay. 

## 2. Results and Discussion

### 2.1. Isolation and Spectroscopic Analyses of the Compounds

The 1-BuOH fractions of methanolic extract of Lasianthus verticillatus (Lour.) Merr. leaves were separated by solvent fractionation, various column chromatographies (CC), and HPLC to afford compounds **1**–**8** (Figure 1). The chemical structures of the isolated compounds were investigated and identified through intensive spectroscopic analyses based on UV-visible, one-dimensional (1D) and two-dimensional (2D) NMR, and high-resolution electrospray ionization mass spectra (HR-ESI-MS) as follows (Appendix A).

#### 2.1.1. Chemical Structure of Lasianoside A

Lasianoside A (**1**), [α]^22^_D_–20.6, was obtained as viscous colorless syrup. Its molecular formula was determined as C_16_H_24_O_10_ based on a sodium adduct ion at *m/z* 399.1263 [M + Na]^+^ (calcd for C_16_H_24_O_10_Na 399.1262) in HR-ESI-MS, which suggests five degrees of unsaturation. The IR spectrum exhibited absorption bands at 3390 and 1748 cm^−1^, which indicated the presence of hydroxy and a five-membered lactone ring, respectively. ^1^H-NMR spectrum displayed signals of four methylenes, including a methylene at δ_H_ 2.75 (br d, *J =* 18.1 Hz) and 2.95 (dd, *J =* 18.1, 6.3 Hz), and three methylenes attached to oxygen at δ_H_ 3.91 (2H, d, *J =* 3.9 Hz), 4.20 (br d, *J =* 13.4 Hz), 4.24 (br d, *J =* 13.4 Hz), 4.28 (br d, *J =* 12.0 Hz), and 4.61 (br d, *J =* 12.0 Hz) (Table 1). Furthermore, the ^1^H-NMR data supported the existence of three methines; two of them showed signals at δ_H_ 2.99 (m) and 3.73 (d, *J =* 6.3 Hz), and an oxygenated methine proton at δ_H_ 5.14 (t, *J =* 6.3 Hz).

Taking into consideration the above-mentioned molecular formula and degree of unsaturation, the 16 carbon signals in the ^13^C-NMR spectrum are thus ascribable to the presence of three oxymethylenes (δc 58.4, 63.3, 64.4), an oxymethine (δc 83.2), as well as two unsaturated quaternary carbons (δc 134.4, 140.2) and carbonyl carbon (δc 180.9) (Table 2). The COSY (correlation spectroscopy) (Figure 2) correlations between H_2_-3/H-4 and H-4/H-5 together with the HMBC (heteronuclear multiple bond connectivity correlation) (Figure 2) correlations from H-5 to C-11 (δc 180.9), C-3 (δc 63.3), C-8 (δc 140.2), and C-9 (δc 134.4) and from H-6 to C-11 (δc 180.9) indicated that **1** shared the same core structure as **6** [11]. Moreover, the 1D NMR data of **1** also gave an anomeric proton signal at δ_H_ 4.30 (d, *J =* 7.9 Hz), in addition to six carbon signals at δ_c_ 103.8, 78.0, 78.0, 75.1, 71.6, and 62.8 that belong to a glucose moiety, which means that **1** was the glucoside of **6**. The HMBC correlation from the anomeric proton of glucose H-1′ (δ_H_ 4.30) to C-1 (δc 64.4) of the aglycone moiety indicated the glucosylation position on C-1. The presence of a D-glucose moiety was confirmed by the acid hydrolysis of **1** followed by HPLC analysis with a chiral detector. The coupling constant (*J =* 7.9 Hz) of H-1′ indicated β linkage for glucose moiety. The relative configuration was established by NOESY analysis (Figure 3). The correlation between H-3/H-5, H-5/H-6, and H-6/H-7 suggested a β-orientation, while the absence of correlation signals between H-4 and H-6 suggested a α-orientation. The chemical shift values and the coupling patterns of aglycone moiety were essentially superimposable to **6** except for the glucosylation position C-1, which supported the relative stereochemistry of **1**, as shown in Figure 1. The absolute stereochemistry of **1** was elucidated by comparison of the circular dichroism (CD) spectrum. The positive cotton effect at 221 nm (∆ε = + 0.854) showed the same absolute stereochemistry with lasianol (**6**) [11], and consequently the structure of **1** was determined to be a (4*R*, 5*R*, 6*S*) configuration, i.e., 1-*O*-β-d-glucopyranoside of **6**.

#### 2.1.2. Chemical Structure of Lasianoside B

Lasianoside B (**2**), [α]^22^_D_–26.6, was isolated as viscous colorless syrup and displayed the same molecular formula C_16_H_24_O_10_ of **1** in HR-ESI-MS at *m/z* 399.1265 [M + Na]^+^ (calcd for C_16_H_24_O_10_ Na 399.1262). Analysis of 1D and 2D NMR data (Table 1 and Table 2) revealed that the data of **2** were very similar to those of **1** except for apparent differences for the signals of H_2_-1 [δ_H_ 4.16 (br d, *J =* 13.0 Hz), 4.33 (br d, *J =* 13.0 Hz)] and H_2_-3 [δ_H_ 3.93 (dd, *J =* 9.7, 5.5 Hz) and 4.23 (dd, *J =* 9.7, 4.0 Hz)] in **2** compared to the corresponding signals of H_2_-1 and H_2_-3 in **1**. The downfield shift of H_2_-3 (δ_H_ 3.93 and 4.23) suggested that the glucose moiety in **2** was linked at C-3 instead of C-1 as in **1**. This assumption was confirmed by HMBC correlation from anomeric proton H-1′ (δ_H_ 4.33) to C-3 (δc 70.5) (Figure 2). The relative and absolute configurations of **2** were determined to be the same as those of **1** by NOESY (Figure 3) and CD spectral data; thus, the structure of **2** was deduced to be 3-*O*-β-d-glucopyranoside of **6**.

#### 2.1.3. Chemical Structure of Lasianoside C

Lasianoside C (**3**), [α]^22^_D_–10.2, was obtained as viscous pale yellow syrup; it had a molecular formula of C_25_H_30_O_13_ with 11 degrees of unsaturation as determined by HR-ESI-MS (measured at *m/z*: 561.1572 [M+Na]^+^, calcd for C_25_H_30_O_13_Na: 561.1579). The UV spectrum showed absorption maxima at 329, 300, and 245 nm, suggesting the presence of an aromatic ring. In the same way, the IR spectrum of **3** exhibited absorption bands at 3356, 1745, 1712, 1623, 1600, and 1512 cm^−1^, which suggested the presence of hydroxy, a five-membered lactone ring, conjugated ester carbonyl, α,β-unsaturated olefinic carbons, and an aromatic ring, respectively. The ^13^C-NMR spectrum closely resembled that of **2**, except for the presence of an additional eight sp^2^ signals (δc 114.8, 115.2, 116.5, 123.1, 127.6, 146.8, 147.3, and 149.7) together with a carboxyl carbon signal (δc 169.1) (Table 2). Thus, compound **3** was expected to be an acylated derivative of **2**. Moreover, the ^1^H-NMR spectrum of **3** showed three ABX-coupled aromatic protons, H-2′′ at δ_H_ 7.07 (d, *J =* 1.7 Hz), H-5′′ at δ_H_ 6.80 (d, *J =* 8.2 Hz), and H-6′′ at δ_H_ 6.98 (dd, *J =* 8.2, 1.7 Hz), and two trans-olefinic protons, H-7′′ at δ_H_ 7.57 (br d, *J =* 15.9 Hz) and H-8′′ at δ_H_ 6.30 (br d, *J =* 15.9 Hz) (Table 1). This data implied that the acyl moiety is a trans-caffeoyl group. The esterification position was indicated to be at C-6′ of the glucose moiety, due to the downfield shift of H_2_-6′ to δ_H_ 4.34 (dd, *J =* 12.0, 5.7 Hz) and 4.57 (dd, *J =* 12.0, 1.8 Hz). This assumption was confirmed by HMBC correlation from H_2_-6′ (δ_H_ 4.34 and 4.57) to carbonyl carbon C-9′′ (δc 169.1), while the glucosylation position was indicated to be at C-3 due to HMBC correlation from anomeric proton H-1′ (δ_H_ 4.37) to C-3 (δc 71.2) (Figure 2). The acid hydrolysis of **3** released d-glucose, while alkaline hydrolysis afforded caffeic acid. These results were identified by HPLC comparing with authentic samples. The configuration of glucopyranose was assigned to be β according to the coupling constant of the anomeric proton H-1′ at δ_H_ 4.37 (d, *J =* 8.0 Hz), while the caffeoyl configuration was determined as an E configuration according to the coupling constant of the olefinic protons H-7′′ and H-8′′ (*J =* 15.9 Hz). The relative and absolute configurations of **3** were determined by comparing its NOESY correlation (Figure 3) and CD spectrum to those of **2**. Therefore, the structure of **3** was established to be 6′-caffeoyl-3-*O*-β-d-glucopyranoside of **6**, which was designated as lasianoside C.

#### 2.1.4. Chemical Structure of Lasianoside D

Lasianoside D (**4**), [α]^22^_D_–22.9, was also obtained as viscous pale yellow syrup with the molecular formula C_27_H_32_O_14_ with 12 degrees of unsaturation, which was deduced by the molecular ion peak [M+Na]^+^ at *m/z*: 603.1685 (calcd for C_27_H_32_O_14_Na: 603.1684) in HR-ESI-MS. Analysis of ^1^H and ^13^C-NMR (Table 1 and Table 2), together with the molecular formula revealed that compound **4** was an acetyl derivative of **3**. In the HMBC spectrum (Figure 2), downfield-shifted H_2_-1 (δ_H_ 4.75) showed correlation to the acetyl carbonyl carbon at δc 172.6, which indicated that **4** should be a 1-acetyl derivative of **3**. Further analyses of NMR data suggested that the other part of **4** was the same as that of **3**. The relative and absolute configurations of **4** were also identified to be the same as those of **3** by comparing the NOESY (Figure 3) and CD spectral data with those of **3**; thus, the structure of **4** was elucidated to be 1-acetyl-6′-caffeoyl-3-*O*-β-d-glucopyranoside of **6**, and designated as lasianoside D.

#### 2.1.5. Chemical Structure of Lasianoside E

Lasianoside E (**5**), [α]^22^_D_–24.5, has the same molecular formula as that of **4** by HR-ESI-MS (*m/z*: 603.1682 [M + Na]^+^, calcd for C_27_H_32_O_14_Na: 603.1684). A further comparison of NMR data (Table 1 and Table 2) displayed that compounds **5** and **4** shared the same functional group. In HMBC spectrum (Figure 2), a significant downfield shift of H_2_-10 (δ_H_ 4.67) showed correlation to acetyl carbonyl carbon at δc 172.6. Detailed analyses of 2D NMR data indicated that the other part of **5** was the same as of that of **4**. The relative and absolute configurations of **5** were determined by comparison of the NOESY experiment (Figure 3) and CD data with those of **4**. Therefore, the structure of **5** was elucidated to be 10-acetyl-6′-caffeoyl-3-*O*-β-d-glucopyranoside of **6**, and designated as lasianoside E.

### 2.2. Antioxidant and Cytotoxic Activities of Isolated Compounds

Compounds **1**–**8** from the 1-BuOH fraction of *L. verticillatus* were evaluated for their DPPH radical scavenging activities. As shown in Table 3, lasianosides C and D (**3** and **4**) displayed potent radical scavenging activities (IC_50_: 30.2 ± 1.8 and 32.0 ± 1.2 µM, respectively) that were comparable with those of standard Trolox (IC_50_: 29.2 ± 0.39 µM), while compound **5** exhibited moderate scavenging activity (IC_50_: 46.4 ± 2.3 µM). Moreover, the remaining compounds did not possess significant DPPH radical scavenging properties (IC_50_ >100 µM). These results suggested that the radical scavenging activities of **3**–**5** may be related to the presence of caffeoyl moieties in their structures [19,20]. On other hand, none of the isolated compounds exhibited cytotoxic activity against human lung cancer cell line (A549) as shown in Table 3, which coincided with our preliminary result for 1-BuOH extract (IC_50_ > 100 µM).

## 3. Materials and Methods

### 3.1. General Method

UV and IR spectra were obtained on a Jasco V-520 UV/Vis spectrophotometer and a Horiba FT-710 Fourier transform infrared spectrophotometer (Horiba, Kyoto, Japan), respectively. Optical rotations data were measured on a JASCO P-1030, while CD spectra were recorded on a Jasco J-720 circular dichroism spectrometer (Jasco, Tokyo, Japan). NMR experiments were measured on a Bruker AVANCE 500-MHz and 700-MHz spectrometers, with tetramethylsilane (TMS) as an internal standard (Bruker, Billerica, MA, USA). Positive ion HR-ESI-MS spectra were recorded using an LTQ Orbitrap XL mass spectrometer (Thermo Fisher Scientific, Waltham, MA, USA). Column chromatography (CC) was performed on a Diaion HP-20 (Mitsubishi chemical Corp., Japan), silica gel 60 (230–400 mesh, Merck, Germany), and octadecyl silica (ODS) gel (Cosmosil 75C_18_-OPN (Nacalai Tesque, Kyoto, Japan; Φ = 35 mm, *L* = 350 mm), and TLC was performed on precoated silica gel plates 60 GF_254_ (0.25 mm in thickness, Merck). HPLC was performed on ODS gel (Cosmosil 5C_18_-AR, Nacalai Tesque, Kyoto, 10 mm, 250 mm, flow rate 2.5 mL/min) with a mixture of H_2_O, acetone, and MeOH, and the elute was monitored by refractive index and/or a UV detector. An amino column (Shodex Asahipak NH2P-50 4E (4.6 mm × 250 mm), CH_3_CN-H_2_O (3:1) 1mL/min) was used together with a chiral detector (Jasco OR-2090*plus*) for the HPLC analysis of sugars obtained after hydrolysis.

### 3.2. Plant Material

Leaves of *Lasianthus verticillatus* were collected from Iriomote Island, Okinawa Prefecture, Japan, in September 2000. The voucher specimen (IR0009-LT) was deposited at the department of pharmacognosy, faculty of pharmaceutical sciences, Hiroshima university.

### 3.3. Extraction and Isolation

The air-dried and powdered leaves of *L. verticillatus* (Lour.) Merr. (7.0 kg) were extracted by maceration with MeOH (98 L × 2) and concentrated to 90% MeOH solution; then, they were defatted with 3 L of *n*-hexane. The remaining solution was evaporated and re-suspended in 1 L of H_2_O and extracted with 3 L of EtOAc and 3 L of 1-BuOH, successively.

Part of the 1-BuOH fraction (124.5 g) was fractionated on a Diaion HP-20 column (Φ = 10 cm, *L* = 350 cm, 2.5 kg), and eluted with H_2_O (15 L), followed by an MeOH/H_2_O step gradient solvent systems (10%, 20%, 30%, 40%, 60%, and 100% MeOH, 15 L each, 1-L fractions being collected), similar fractions were grouped together to give 20 fractions (Fr. Lt1–Lt20). Fraction Lt8 (18.7 g) was proceeded on silica gel CC (Φ = 4.5 cm, *L* = 50 cm, 400 g), started with CHCl_3_ (2.5 L), and followed by CHCl_3_/MeOH developing solvent systems (7%, 10%, 15%, 20%, 30%, and 100% MeOH, 2.5 L each) to obtain 16 fractions (Fr. Lt8.1–Lt8.16). Each fraction of Lt8.12 (681 mg) and Lt8.13 (240 mg) was subjected to open reversed phase (ODS) CC with 10% aq. methanol (400 mL) to 100% methanol (400 mL), linear gradient, to give (Frs. Lt8.12.1–Lt8.12.6 and Frs. Lt8.13.1–Lt8.13.6, respectively). The residue Lt8.12.2 (131 mg) was purified by HPLC (ODS) with 5% aq. methanol to give **1** (4.0 mg), while residue Lt8.13.2 (174 mg) was purified by HPLC (ODS) with 5% aq. methanol to give **6** (10.4 mg), **2** (5.0 mg). Fraction Lt13 (4.3 g) was proceeded on silica gel CC (Φ = 4 cm, *L* = 40 cm, 230 g), started with CHCl_3_ (1.5 L), and followed by CHCl_3_/MeOH developing solvent systems (5%, 7%, 10%, 15%, 20%, 30%, 40%, and 100% MeOH, 1.5 L each) to give 14 fractions (Frs. Lt13.1–Lt13.14). The residue Lt13.5 (455 mg) was purified by preparative HPLC (ODS) with 15% aq. acetone to give **7** (5.8 mg) and **8** (23.4 mg). The other residue Lt13.10 (215 mg) was purified by preparative HPLC (ODS) with 30% aq. methanol to give **3** (6.0 mg). Fraction Lt15 (7.2 g) was chromatographed on silica gel CC (Φ = 5.2 cm, *L* = 38 cm, 350 g), started with CHCl_3_ (2.5 L), and followed by CHCl_3_/MeOH developing solvent systems (5%, 10%, 12%, 15%, 20%, 30%, and 100% MeOH, 2.5 L each) to obtain 12 fractions (Frs. Lt15.1–Lt15.12). Fraction Lt15.5 (656 mg) was rechromatographed on silica gel CC (Φ = 2.5 cm, *L* = 50 cm, 120 g), started with CHCl_3_ (500 mL), and followed by CHCl_3_/MeOH developing solvent systems (5%, 7%, 10%, 12%, and 100% MeOH, 500 mL each) to give 11 fractions (Frs. Lt15.5.1–Lt15.5.11). Fraction Lt15.7 (577 mg) was separated by HPLC (ODS) with 28% aq. acetone to give **4** (3.0 mg). The fraction LtB17 (6.1 g) was further purified by silica gel column chromatography (Φ = 5 cm, *L* = 40 cm, 380 g), eluting with a stepwise CHCl_3_/MeOH gradient (100:0 to 70:30, 2.4 L each) to obtain 11 fractions (Frs. LtB17.1–LtB17.11). The residue LtB17.7 (839 mg) was separated by HPLC, 28% aq. acetone to obtain **5** (7.0 mg).

### 3.4. Spectroscopic Data of Compounds ***1***–***5***

*Lasianoside A* (**1**) Viscous colorless syrup [*α*]^22^_D_ –20.6 (*c* 0.30, MeOH); HR-ESI-MS: *m/z*: 399.1263 [M+Na]^+^ (calcd for C_16_H_24_O_10_Na, 399.1262); CD *λ*_max_ (*c* 2.66 × 10^−5^ M, MeOH) nm (*∆ɛ*): 270 (–0.059), 221 (+0.854); IR (film) *ν*_max_: 3390, 2921, 1748, 1194, 1072, 1032 cm^−1^; ^1^H and ^13^C data, see Table 1 and Table 2.

*Lasianoside B* (**2**) Viscous colorless syrup [*α*]^22^_D_ –26.6 (*c* 0.30, MeOH); HR-ESI-MS: *m/z*: 399.1265 [M+Na]^+^ (calcd for C_16_H_24_O_10_Na, 399.1262); CD *λ*_max_ (*c* 2.66 × 10^−5^ M, MeOH) nm (*∆ɛ*): 281 (–0.026), 217 (+1.99); IR (film) *ν*_max_: 3372, 2927, 1754, 1171, 1078, 1024 cm^−1^; ^1^H and ^13^C data, see Table 1 and Table 2.

*Lasianoside C* (**3**) Viscous pale yellow syrup [*α*]^22^_D_ –10.2 (*c* 0.34, MeOH); HR-ESI-MS: *m/z*: 561.1572 [M+Na]^+^ (calcd for C_25_H_30_O_13_Na, 561.1579); CD *λ*_max_ (*c* 1.85 × 10^−5^ M, MeOH) nm (*∆ɛ*): 283 (–0.357), 221 (+6.155); UV (MeOH) *λ*_max_ nm (log *ɛ*): 329 (4.74), 300 (4.63), 245 (4.59); IR (film) *ν*_max_: 3356, 2913, 1745, 1712, 1623, 1600, 1512, 1360, 1274, 1165, 1032 cm^−1^; ^1^H and ^13^C data, see Table 1 and Table 2.

*Lasianoside D* (**4**) Viscous pale yellow syrup [*α*]^22^_D_ –22.9 (*c* 0.27, MeOH); HR-ESI-MS: *m/z*: 603.1685 [M+Na]^+^ (calcd for C_27_H_32_O_14_Na, 603.1684); CD *λ*_max_ (*c* 1.40 × 10^−5^ M, MeOH) nm (*∆ɛ*): 276 (–0.105), 208 (+1.010); UV (MeOH) *λ*_max_ nm (log *ɛ*): 329 (4.37), 295 (4.27), 233 (4.42); IR (film) *ν*_max_: 3396, 2920, 1746, 1732, 1713, 1634, 1608, 1518, 1355, 1264, 1163, 1032 cm^−1^; ^1^H and ^13^C data, see Table 1 and Table 2.

*Lasianoside E* (**5**) Viscous pale yellow syrup [*α*]^22^_D_ –24.5 (*c* 0.40, MeOH); HR-ESI-MS: *m/z*: 603.1682 [M+Na]^+^ (calcd for C_16_H_24_O_10_Na, 603.1684); CD *λ*_max_ (*c* 1.72 × 10^−5^ M, MeOH) nm (*∆ɛ*): 281 (–0.297), 214 (+1.324); UV (MeOH) *λ_m_*_ax_ nm (log *ɛ*): 331 (3.98), 292 (3.90), 240 (3.96); IR (film) *ν*_max_: 3361, 2928, 1747, 1738, 1714, 1625, 1602, 1509, 1365, 1271, 1179, 1030 cm^−1^; ^1^H and ^13^C data, see Table 1 and Table 2.

### 3.5. Acid Hydrolysis

The isolated compounds **1**–**5** (1.0 mg each) were hydrolyzed with 1 M HCl (1.0 mL) at 80 °C for 3 h. The reaction mixtures were neutralized with amberlite IRA96SB (OH¯ form), and the resin was removed by filtration. Then, the filtrate was extracted with an equal volume of EtOAc. The aqueous layers were analyzed by HPLC with an amino column [Asahipak NH2P-50 4E, CH_3_CN-H_2_O (3:1), 1mL/min] and a chiral detector (JASCO OR-2090*plus*) in comparison with d-glucose as authentic standard. The aqueous layers of hydrolyzed compounds showed a peak at *t*_R_ 8.15 min, which coincided with that of d-glucose [21].

### 3.6. Alkaline Hydrolysis

Compounds **3**, **4**, and **5** (3.0 mg each) were prepared as solutions in 50% aqueous 1,4-dioxane (0.5 mL). These solutions were treated with 10% aqueous KOH (0.5 mL) and stirred at 37 °C for 1 h. The reaction mixtures were neutralized with an ion-exchange resin (Amberlite IR-120B (H^+^-form), and then filtrated and evaporated. The residues were dissolved in EtOAc and subjected to HPLC analysis [Cosmosil C_18_-PAQ, MeOH-H_2_O (45:50), with 0.1% TFA, 1mL/min)] with UV (245 nm) detector to identify the caffeoyl moieties at *t*_R_ 11.42 min [22].

### 3.7. DPPH Radical Scavenging Activity

Free radical scavenging activity was evaluated by using a quantitative DPPH assay. The absorbance of various concentrations of test compounds dissolved in 100 µL of MeOH in a 96-well microtiter plate was measured at 515 nm as (A_blank_). Then, 100 µL of DPPH solution (200 µM) was added to each well. The plate was incubated in a dark chamber at room temperature for 30 min before measuring the absorbance (A_sample_) again. The following equation was used to calculate the percentage inhibition of free radicals:% Inhibition = [1 − (A_sample_ − A_blank_)/(A_control_ − A_blank_)] × 100
where A_control_ is the absorbance of control (DMSO and all reagents, except for the test compound). Trolox was used as the positive control. Their IC_50_ values were determined based on three independent experiments [23].

### 3.8. Cytotoxicity

Human lung cancer cell proliferation assay was carried out by means of colorimetric MTT assay to evaluate both the potential for antitumor agents and the safety against human cells. The cell line (A549) was established through explant culture of lung carcinomatous tissue from a 58-year-old Caucasian male and frequently used for in vitro antitumor drug screening. In this study, we also aimed to elucidate the safety of our antioxidant compounds using this human cell line because of the ease of obtaining and culturing indefinitely rather than normal and primary cultures. The cells were cultured in Dulbecco’s modified Eagle’s medium (DMEM) supplemented with 10% heat-inactivated fetal bovine serum, amphotericin B (5.6 µg/mL), and kanamycin (100 µg/mL). In a 96-well plate, the various concentrations of test compounds were incubated with A549 cells (5 × 10^3^ cells/well) in a 5% CO_2_ incubator at 37 °C for 72 h. Then, the cell shape, approximate number, and color of supernatants were checked by microscope to avoid an over or underestimation of the result obtained by the following colorimetric evaluation. Then, an MTT solution (0.5 mg/mL) was replaced in each well, and the plate was further incubated for 1.5 h to generate formazan crystals. Subsequently, the medium was completely removed; then, 100 µL of DMSO was added to each well to dissolve the crystals. The optical density (OD) values were measured at 540 nm using a microplate reader. The percentage (%) inhibition of cell growth was calculated using the following equation:% Inhibition = [1 − (A_sample_ − A_blank_)/(A_control_ − A_blank_)] × 100
where A_control_ is the absorbance of control (DMSO and all reagents, except for the test compound). Their IC_50_ values were determined based on three independent experiments [24].

## 4. Conclusions

In this study, the chemical composition of *L. verticillatus* leaves was studied for the first time and resulted in the isolation of five undescribed iridoid glucosides, lasianosides A–E (**1**–**5**), together with three known compounds (**6**–**8**). The chemical structures of the new compounds were elucidated on the basis of NMR and MS studies. The absolute configuration of the new compounds was determined by the CD method. The isolated five new compounds (**1**–**5**) have the same core structure lasianol (**6**), which is an iridane skeleton. It was first isolated from *Lasianthus wallichii* by our previous research [11]. Lasianol and its derivatives have never been isolated afterward until our present work. The uniqueness of the structure may be produced through the biosynthetic pathway depicted in Figure 4. The branching point is the form of intermediate iridotrial, e.g., hemiacetal and keto forms for iridoid and the iridane skeleton, respectively.

Plant species are potentially good sources for antioxidant secondary metabolites because of their habitats, being exposed continuously to sunlight, and thus gaining protection mechanisms against reactive oxygen species produced by UV irradiation [25]. The isolated compounds were tested for their antioxidant activities by DPPH free radical scavenging assay, and cytotoxicity against human lung cancer cell line (A549) by MTT assay. Among them, compounds **3** and **4** exhibited potent radical scavenging activity, while compound **5** showed moderate scavenging activity. The free radical scavenging activity of **3**, **4**, and **5** may be related to the presence of caffeoyl moieties in their structures. The possible reaction mechanism with reactive oxygen species is shown in Figure 4 according to the mechanism of catechol [26]. It is noteworthy that none of the isolated compounds exhibited cytotoxic activity against human lung cancer cell line A549 by the MTT method. Therefore, compounds **3**, **4**, and **5** have the potential to be safe antioxidant reagents to ameliorate a variety of oxidative stress-related diseases.

## Figures and Tables

**Figure 1 molecules-24-03995-f001:**
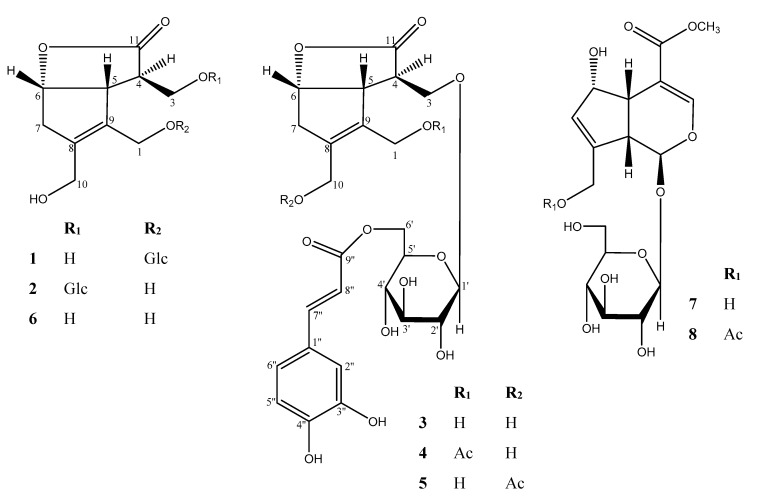
Isolated compounds from *Lasianthus verticillatus* (**1**–**8**).

**Figure 2 molecules-24-03995-f002:**
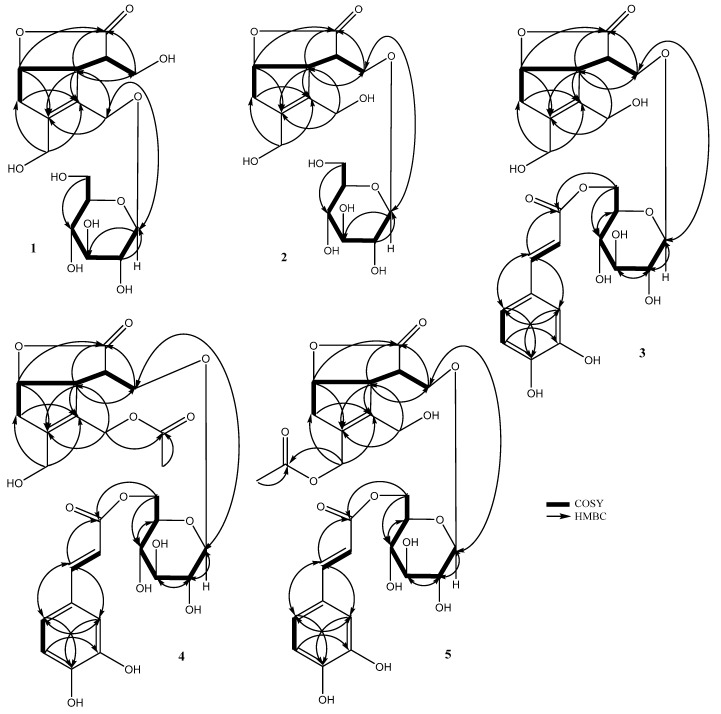
Correlation spectroscopy (COSY) and heteronuclear multiple bond connectivity correlation (HMBC) correlations of **1**–**5**.

**Figure 3 molecules-24-03995-f003:**
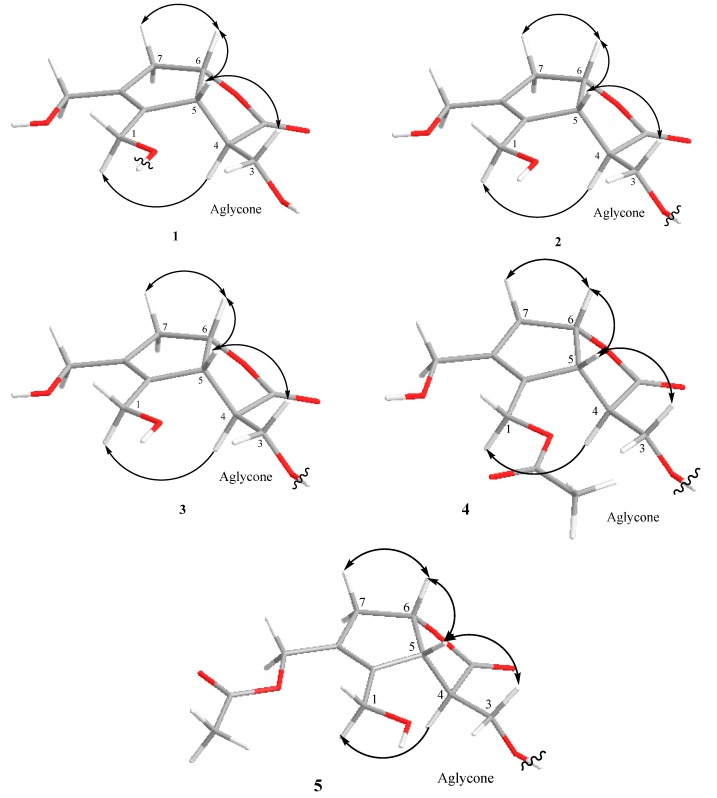
Key NOESY correlations of **1**–**5**.

**Figure 4 molecules-24-03995-f004:**
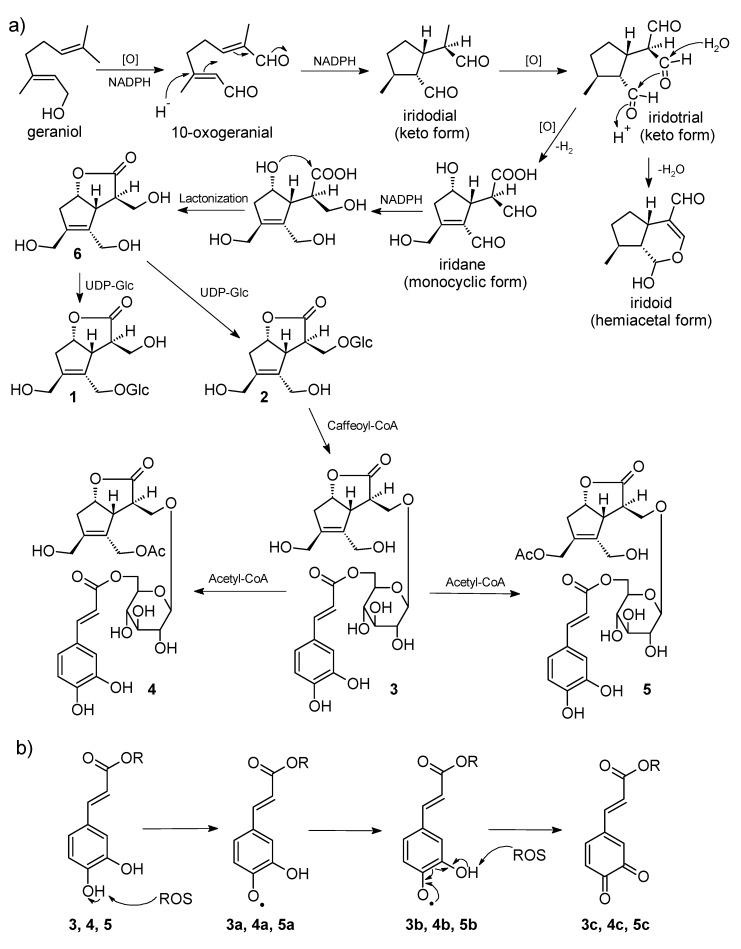
Plausible biosynthetic pathway of compounds **1**–**5**, and possible antioxidant mechanism of compounds **3**–**5**. (**a**) Plausible biosynthetic pathway. Iridotrial (keto form) is a common substrate for the production of iridane and iridoid skeletons. [O]; oxidation. (**b**) Possible antioxidant mechanism. The conjugated para-OH is the expected primary target of reactive oxygen species (ROS) to generate phenoxy radials **3**–**5a**. The second ROS attack results in the formation of ortho quinones, **3**–**5c**, through the resonance structures **3**–**5b**.

**Table 1 molecules-24-03995-t001:** ^1^H NMR data of compounds **1**–**5** (500 MHz, CD_3_OD, δ in ppm, J in Hz).

Position	1	2	3	4	5
1	4.28 br d (12.0) 4.61 br d (12.0)	4.16 br d (13.0) 4.33 br d (13.0)	4.16 br d (13.0) 4.27 br d (13.0)	4.75 br s	4.18 br d (13.4) 4.28 br d (13.4)
3	3.91 2H d (3.9)	3.93 dd (9.7, 5.5) 4.23 dd (9.7, 4.0)	4.02 dd (10.0, 5.0) 4.13 m	4.04 dd (10.1, 3.5) 4.09 dd (10.1, 4.4)	4.04 dd (10.0, 5.0) 4.13 dd (10.0, 3.4)
4	2.99 m	3.00 m	2.98 m	2.94 m	2.99 m
5	3.73 br d (6.3)	3.78 br d (6.3)	3.75 m	3.68 br d (6.3)	3.76 m
6	5.14 t (6.3)	5.21 td (6.3, 1.0)	5.16 t (6.4)	5.13 t (6.3)	5.15 t (6.4)
7	2.75 br d (18.1) 2.95 dd (18.1, 6.3)	2.73 br d (18.1) 2.94 dd (18.1, 6.3)	2.69 br d (18.0) 2.85 dd (18.0, 6.4)	2.69 br d (18.3) 2.82 dd (18.3, 6.3)	2.62 br d (18.0) 2.82 dd (18.0, 6.4)
8	-	-	-	-	-
9	-	-	-	-	-
10	4.20 br d (13.4) 4.24 br d (13.4)	4.20 2H, br s	4.13 2H, br s	4.11 br d (13.7) 4.19 br d (13.7)	4.67 2H, br s
11	-	-	-	-	-
12				-	-
13				2.04 3H, s	2.03 3H, s
1′	4.30 d (7.9)	4.33 d (7.9)	4.37 d (8.0)	4.36 d (8.0)	4.63 d (7.9)
2′	3.18 dd (9, 7.9)	3.21 dd (9.1, 7.9)	3.23 t (8.0)	3.24 dd (9.1, 8.0)	3.23 dd (9.0, 7.9)
3′	3.37 t (8.8)	3.38 t (9.1)	3.40 m	3.40 m	3.40 m
4′	3.28 m	3.30 m	3.39 m	3.39 m	3.39 m
5′	3.29 m	3.31 m	3.56 m	3.56 m	3.56 m
6′	3.68 dd (12, 5.5) 3.90 dd (12, 1.8)	3.72 dd (12, 5.2) 3.89 dd (12, 1.3)	4.34 dd (12.0, 5.7) 4.57 dd (12.0, 1.8)	4.43 dd (12.0, 5.7) 4.59 dd (12.0, 2.0)	4.35 dd (11.9, 6.0) 4.56 dd (11.9, 2.0)
1″			-	-	-
2″			7.07 d (1.7)	7.05 d (2.0)	7.06 d (2.0)
3″			-	-	-
4″			-	-	-
5″			6.80 d (8.2)	6.80 d (8.2)	6.80 d (8.2)
6″			6.98 dd (8.2, 1.7)	6.97 dd (8.2, 2.0)	6.97 dd (8.2, 2.0)
7″			7.57 br d (15.9)	7.56 br d (15.9)	7.56 br d (15.9)
8″			6.30 br d (15.9)	6.28 br d (15.9)	6.30 br d (15.9)
9″			-	-	-

m: multiplet or overlapped signals.

**Table 2 molecules-24-03995-t002:** ^13^C-NMR data of compounds **1**–**5** (175 MHz, CD_3_OD, δ in ppm).

Position	1	2	3	4	5
1	64.4	56.8	58.4	59.4	57.1
3	63.3	70.5	71.2	71.2	71.3
4	48.4	46.8	46.9	47.0	46.8
5	53.4	53.0	53.4	53.4	53.3
6	83.2	83.3	83.4	83.3	83.2
7	41.8	41.7	41.6	41.5	41.7
8	140.2	138.5	138.5	142.0	133.4
9	134.4	137.4	137.4	132.5	140.5
10	58.4	58.4	57.0	58.5	60.9
11	180.9	180.2	180.2	180.0	180.0
12				172.6	172.6
13				20.8	20.7
1′	103.8	104.6	104.8	104.8	104.9
2′	75.1	75.0	75.0	75.0	75.0
3′	78.0	77.9	77.7	77.8	77.7
4′	71.6	71.5	71.6	71.6	71.6
5′	78.0	78.1	75.5	75.5	75.5
6′	62.8	62.7	64.3	64.2	64.3
1″			127.6	127.6	127.6
2″			115.2	115.2	115.2
3″			146.8	146.9	146.9
4″			149.7	149.8	149.7
5″			116.5	116.5	116.5
6″			123.1	123.1	123.1
7″			147.3	147.3	147.3
8″			114.8	114.8	114.8
9″			169.1	169.0	169.1

**Table 3 molecules-24-03995-t003:** Bioactivities of isolated compounds **1**–**8** (**n** = **3**) *****. DPPH: 1,1-diphenyl-2-picrylhydrazyl.

Isolated Compounds	DPPH (IC_50_ µM)	A549 Cytotoxicity (IC_50_ µM)
Lasianoside A (**1**)	>100	>100
Lasianoside B (**2**)	>100	>100
Lasianoside C (**3**)	30.2 ± 1.8	>100
Lasianoside D (**4**)	32.0 ± 1.2	>100
Lasianoside E (**5**)	46.4 ± 2.3	>100
Lasianol (**6**)	>100	>100
Deacetyl daphylloside (**7**)	>100	>100
Daphylloside (**8**)	>100	>100
Trolox	29.2 ± 0.39	-
Doxorubicin	-	0.90 ± 0.02

* The results were obtained from three independent triplicate experiments, and the IC_50_ values were calculated by linear regression on Excel software.

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
