# Peer review of "Lasianosides A–E: New Iridoid Glucosides from the Leaves of Lasianthus verticillatus (Lour.) Merr. and Their Antioxidant Activity"

_molecules, 2019, doi:10.3390/molecules24213995_

Round 1
Reviewer 1 Report
This paper by Gadah Al-Hamoud et al identified five new compounds and three known compounds from the 1-BuOH soluble fraction of aerial parts of Lasianthus verticillatus. The general purpose of this study is clear, and the study appears to be of interest. However, I recommend you to add more information about Method. Overall I recommend this paper to be published in Molecules after addressing the following comments.
Major Point
Introduction
The authors clearly show that all of the isolated compounds have low cytotoxicity in human lung cancer line cell A549. I recommend you to add more information about A549 cells in the introduction section. Moreover you should describe the reason why did you chose this cell line. Why did you use normal cell or primary cultured cells? The authors should explain this point in the revised manuscript.
Method
MTT assay is commonly used to assess the cell survival. However, there have been some reports insisting that MTT assay exhibited non-specific intracellular reduction of tetrazolium which led to underestimated results of cytotoxicity.
Is there a possibility that the isolated compounds directly interact with interact directly with MTT? Therefore, I recommend you to add more data about the effect of isolated compounds on MTT value in the cell free condition. Otherwise, the authors should explain this point in the revised manuscript.
Mainor point
Please indicate the number of each experiments. Especially, MTT assay and DPPH assay.
Please add the concentration of MTT in the method section.
Author Response
This paper by Gadah Al-Hamoud et al identified five new compounds and three known compounds from the 1-BuOH soluble fraction of aerial parts of Lasianthus verticillatus. The general purpose of this study is clear, and the study appears to be of interest. However, I recommend you to add more information about Method. Overall I recommend this paper to be published in Molecules after addressing the following comments.
Thank you for the very kind comments. We revised as follows.
Major Point
Introduction:
The authors clearly show that all of the isolated compounds have low cytotoxicity in human lung cancer line cell A549. I recommend you to add more information about A549 cells in the introduction section.
Thank you for the very useful comments. We added the description on lung cancer and this cell line in the introduction and method sections
Moreover you should describe the reason why did you chose this cell line. Why did you use normal cell or primary cultured cells? The authors should explain this point in the revised manuscript.
Thank you for the very important comments. Our initial objectives were discovery of bioactive compounds such as antioxidant and antitumor agents. However the cytotoxicity assay essentially includes the two different viewpoints such as cytotoxicity for antitumor agent and safety for human use. Therefore we used human tumor cell line (A549) for both aims.
In actually, we only claimed the aspect of safety of our antioxidant compounds in results and discussion section according to the result of cytotoxicity assay.
The other reason for the use of A549 rather than normal and primary cultures is easiness to culture definitely and was documented in method section.
Method
MTT assay is commonly used to assess the cell survival. However, there have been some reports insisting that MTT assay exhibited non-specific intracellular reduction of tetrazolium which led to underestimated results of cytotoxicity. Is there a possibility that the isolated compounds directly interact with interact directly with MTT? Therefore, I recommend you to add more data about the effect of isolated compounds on MTT value in the cell free condition. Otherwise, the authors should explain this point in the revised manuscript.
Thank you for informing us such important studies. In my opinion, there are various methods to measure cytotoxicity, and all of these methods potentially have advantage and disadvantage by comparing with each other. Therefore MTT method is still one of the gold standard. However, I need to keep this possibility in my mind to avoid misunderstanding of our results.
We usually check the cell shape and approximate number by microscope to avoid the over or under estimation of the following MTT reaction. I hope this procedure is essentially enough to trust our results. And some description was added in Method section on this issue.
If we find the discrepancy between the microscopic observation and MTT color, we should try to find better method, e.g. neutral red uptake (NRU), resazurin reduction (RES) and sulforhodamine B (SRB) assays etc., for the specific compounds one by one.
Mainor point
Please indicate the number of each experiments. Especially, MTT assay and DPPH assay.
Thank you for the useful comment.
n=3 was added in Table 3 and Method section.
Please add the concentration of MTT in the method section.
Thank you for the useful comment.
0.5mg/ml was added in Method section
Reviewer 2 Report
This paper attempts to prove the antioxidant and cytotoxic potential of L. verticillatus leaves extract and their phytochemical investigation.
Overall, the methodology used could be improved, and the section of results and discussion must be meliorate. The manuscript feet the journal’s scope.
Comments or suggestions:
Title is vague and unclear and does not reflect the content of the manuscript. Authors should correct it.
The abstract only describes the methodology. It is very incomplete, it does not refer to the introduction, the results and discussion or conclusions.
The keywords repeat the words of the title. Avoid using the same title words as this will increase the probability that the article will be detected in a search. So, I suggest that authors correct the keywords.
The introduction is not well enough described to highlights all the subjects needed to understand all the manuscript. It's clear but not enough detailed.
Line 36: The authors speak of the genus Lasianthus, but do not even mention that it is a plant.
Results and discussion
Results data is organized in an orderly and logical sequence.
The most relevant results are highlighted.
The discussion not explains the data presented in the results, by comparing with recent work.
The results do not refer to the cytotoxic potential of the extracts? It must be explain and discussed.
Author Response
This paper attempts to prove the antioxidant and cytotoxic potential of L. verticillatus leaves extract and their phytochemical investigation. Overall, the methodology used could be improved, and the section of results and discussion must be meliorate. The manuscript feet the journal’s scope.
Thank you for the careful review and comments
Comments or suggestions:
Title is vague and unclear and does not reflect the content of the manuscript. Authors should correct it.
Thank you for the helpful comment.
The title was changed according to the reviewer's comment.
The abstract only describes the methodology. It is very incomplete, it does not refer to the introduction, the results and discussion or conclusions.
Thank you for the useful comment.
The Abstract was changed according to the reviewer's comment.
The keywords repeat the words of the title. Avoid using the same title words as this will increase the probability that the article will be detected in a search. So, I suggest that authors correct the keywords.
Thank you for the useful comment.
The keywords were changed according to the reviewer's comment.
The introduction is not well enough described to highlights all the subjects needed to understand all the manuscript. It's clear but not enough detailed.
Thank you for important comment.
The introduction was changed according to the reviewer's comment.
Line 36: The authors speak of the genus Lasianthus, but do not even mention that it is a
plant.
Very sorry for our carelessness. The word "Plants" was added at the corresponding position according to the reviewer's comment.
Results and discussion
Results data is organized in an orderly and logical sequence.
The most relevant results are highlighted.
Thank you for the careful review and comments
The discussion not explains the data presented in the results, by comparing with recent work.
Thank you for the suggestion of the recent work. We couldn't find the preceding phytochemical studies on this plant at the moment when we prepared the manuscript. However, a recent work has just published in October, 2019. We cited this paper and added description by comparing with recent work according to the reviewer's comment.
The results do not refer to the cytotoxic potential of the extracts? It must be explain and discussed.
Thank you for the useful comment. The 1-BuOH fraction didn't show any significant activity against A549 at the highest concentration of our procedure (100ug/ml). We added the description in the result section.
Plant extract usually contains a huge variety of chemical constituents at first, which may mask the active principles even if they exist. We usually check the bioactivity of the isolated compounds again, even though the initial fraction didn't show any activities.
Round 2
Reviewer 2 Report
I thank the authors for their efforts to significantly improve the manuscript.
I believe that this manuscript shows improvements compared to the previous one. But my main concern remain is related to discussion. The discussion is better written and better supported by bibliographic references but I really think it could be better.
The table should have a footnote, where the authors explain that those results derived from three independent assays and what was the statistics used.
Line 255: “2.2 biological activity of isolated compounds”
Line 265: “The antioxidant activity…” ???
Author Response
Overall, Thank you again for providing us to improve our manuscript.We revised again according to the very useful reviewer's suggestion. The revising points are indicated in green.
Reviewer2 comments
I thank the authors for their efforts to significantly
improve the manuscript.
I believe that this manuscript shows improvements
compared to the previous one. But my main
concern remain is related to discussion. The
discussion is better written and better supported by
bibliographic references but I really think it could be
better.
--- Thank you for the comments, we investigated the preceding papers and added description on the structural uniquness of our compounds and plausible biosynthetic mechanism to help readers understand the difference between iridoid and our compounds, iridane. In addition, the possible mechanism on antioxidant activity was also added to support our result.
The table should have a footnote, where the
authors explain that those results derived from three
independent assays and what was the statistics
used.
---- Thank you for the comment. We added the description.
Line 255: “2.2 biological activity of isolated
compounds”
--- We changed the heading to indicate precisely.
Line 265: “The antioxidant activity…” ???
---- Very sorry, it's our careless mistake. We corrected.
